



# A New Method Towards More Rational Drought Propagation Characterization in Karst Regions

Han Tang[1], Peng Shi[1], Simin Qu[1], Xiaoqing Yang[2,3], Peng Jiang[1], Lanlan Zhao[4], Qiongfang Li[1,2] and Yiqun Sun[1]

[1]Collage of Hydrology and Water Resources, Hohai University, Nanjing 210098, China.

[2]Yangtze Institute for Conservation and Development, Hohai University, Nanjing 210098, China.

[3]Key Laboratory of Hydrologic-Cycle and Hydrodynamic-System of Ministry of Water Resources, Hohai University, Nanjing, China.

[4]Information Center, the Ministry of Water Resources, Beijing 100053, China.

*Correspondence to*: Simin Qu (wanily@hhu.edu.cn)

**Abstract.** Drought propagation time ($T_P$) and lag time ($T_L$) from meteorological to hydrological droughts are two important indicators for characterizing drought propagation, and thus, reasonable estimates of these indicators are conducive to improve the prediction of hydrological droughts. However, traditional quantification methods are mostly based on moving correlations of whole hydro-meteorological series, including the misinformative non-drought periods. Particularly for the karst

regions, the flashy regime of rainfall-runoff relationship during non-drought periods might strongly bias the estimations of the drought propagation indicators. In this study, we propose a new method that use only the informative drought-period data to better quantify the above indicators. Moreover, we compared the effectiveness between the new and traditional methods in regions with different karstification degrees. The results revealed that: (1) hydrological drought events generally exhibited reduced frequencies, longer durations, and smaller deficit volumes than the corresponding meteorological drought events; (2)

compared to conventional methods, the $T_P$ and $T_L$ obtained by using the new method better meet the practical requirements for monitoring more hydrological drought deficit volume, especially in karst regions; (3) the karstification degree is a key factor influencing  the difference between the results obtained using the new and traditional methods, and the superiority of the new method is more pronounced with stronger karstification degrees of the region. The above results can improve our understanding of the drought propagation features in karst regions and can provide an evidence-base for precautional drought-mitigation

actions

## 1 Introduction

Drought has been listed as one of the utmost urgent issues by the Intergovernmental Panel on Climate Change (IPCC), because of the great threat to the world's economy, environment, and social stability (Calvin et al., 2023). The drought hazards are often characterized by wide ranges, high frequencies, and long durations (Dai, 2010; Sheffield et al., 2012; Trenberth et al.,

2013; Yuan et al., 2023), occurring also in humid regions (Lin et al., 2015; Shi et al., 2022; Van Lanen et al., 2013).





Droughts are categorized into different types such as meteorological drought, hydrological drought, agricultural drought, and socio-economic drought (Li et al., 2020; Quiring and Ganesh, 2010), and their transitional linkages in hydrological cycling system are referred to as drought propagation (Eltahir and Yeh, 1999; Peters, 2003; Zhang et al., 2022). The most initiated and essential topic is the propagation from meteorological to hydrological drought (Wu et al., 2018), because of their important

implications for drought early warning and water resources management. Corresponding indicators include response time scale (also termed drought propagation time) (Barker et al., 2016; Ding et al., 2021; Xu et al., 2019), and lag time (Hisdal and Tallaksen, 2003) . The former reflects the accumulative time of meteorological drought to hydrological drought, whereas the latter reflects the time difference between hydrological droughts and the corresponding meteorological droughts (usually indicated by the time lag between their onsets). These two indicators can well represent the temporal propagation

characteristics.

The correlation between meteorological and hydrological droughts is relatively stable compared to the sensitivity of meteorological or hydrological droughts to climate warming (Gu et al., 2021; Lin et al., 2023). Studies have been shown that the drought propagation characteristics can be practically useful to predict the hydrological drought based on the meteorological drought (Lin et al., 2023). Traditional physical hydrological drought prediction, climate models and

hydrological models are often combined to predict runoff, thus realizing the prediction of hydrological drought. However, in some regions (e.g. karst regions, and frozen soil) where traditional hydrological models are less effective, the reliability of those models is naturally affected(Gao et al., 2022; Hartmann et al., 2014; Hu et al., 2022; Meng et al., 2012). Exploring the applicability of drought propagation for hydrological prediction in those regions and its advantages and disadvantages compared to other regions is a primary issue.

Several time-series indices are available to identify drought events, and they can be generally categorized as standardized index and threshold-based index (Van Loon, 2015). The standardized index is calculated by fitting a distribution to the original time series, and it represents anomalies in relative values. In contrasts, the philosophy of the threshold-based index is to maximumly maintain the original time series, representing anomalies in absolute values(Fleig et al., 2006; Zelenhasić and Salvai, 1987). The two categories are both widely used for characterizing drought propagation. For instance, (Lorenzo-Lacruz

et al., 2013) used the standardized streamflow index (SSI) and standardized precipitation index (SPI) to determine the response time; Zhang et al. (2022b). used SPI and standardized runoff index (SRI) to obtain the lag times in the Luanhe River Basin; (Choi et al., 2021) used threshold-based index to analyze the synchrony between meteorological and hydrological droughts in Wisconsin. However, current studies mostly use the whole hydro-meteorological sequences containing both drought and non-drought periods. We argue that the obtained propagation features reflect the general response of the watershed hydrology (i.e.,

discharge dynamics) to the meteorological process (Vicente-Serrano and López-Moreno, 2005; Wu et al., 2021), while not specific responses under droughts, which represent a non-stationary condition of watershed functioning (Yang et al., 2021) . Therefore, it is necessary to further exam the information content of the drought index and to obtain more rational drought propagation features, e.g., using only the informative drought periods of the data series.





Here, we proposed a drought-period method to obtain the propagation times and the lag times of drought propagation based
only on the drought periods of the whole hydro-meteorological time series. Furthermore, we compared its effectiveness to
traditional full-sequence method in the Wujiang River Basin (WJB) in Guizhou Province, China. This region is world-wide
known for its prominent karst features. Additionally, several non-karst watersheds in the region were also used for a thorough
test of the new method under various watershed features. The objectives of this study are: 1) to obtain and compare the $T_P$ and
$T_L$ in study regions using the new drought-period method and traditional full-sequence method; 2) to evaluate the improved
effectiveness of the new method in karst and non-karst regions; 3) to investigate the effect of the karstification degree on the
results.

## 2 Study area and data

### 2.1 Study area

The study area is located mainly in Guizhou Province, in Southwestern China, one of the three largest karst-dominant areas in
the world (Table 1, Fig. 1). The Wujiang River is the largest tributary on the south bank of the upper reaches of the Yangtze
River. The WJB has a typical subtropical humid monsoon climate, and the annual precipitation and mean temperature are
1137.8 mm and 16.0 ℃, respectively. The basin's geomorphology is characterized by mountains, hills, and basins. Carbonate
rock covers approximately 72% of the study area, with karst extensively developed in the basin. Three hydrological stations,
Yachi River, Sinan and Wulong, are considered as control sites, dividing the whole basin into upper (UWJB), middle (MWJB)
and lower (LWJB) watersheds, with drainage areas of 18,180 km$^2$, 33,090 km$^2$ and 31,765 km$^2$, respectively. The karstification
degree changes from strong in UWJB to weak in LWJB. The Qixingguan watershed (QXGW) is a sub-watershed of the UWJB
with a strong karstification degree and an area of 2,999 km$^2$; the Dongtou watershed (DTW) is a sub-watershed of the MWJB
with a moderate karstification degree and an area of 6,197 km$^2$. The Yongwei watershed (YWW, with an area of 13,045 km$^2$)
and the Tongdao watershed (TDW, with an area of 3,784 km$^2$) are two watersheds located in southeastern Guizhou, where
there is no karst in the region.

**Table 1. The information of basins for the study area**

| Basins | Hydrological stations | Area(km$^2$) | Average annual precipitation (mm) | Average annual runoff(mm) | Karstification degree |
|---|---|---|---|---|---|
| UWJB | Yachi River | 18,180 | 1,014.6 | 571.7 | Strong |
| MWJB | Sinan | 33,090 | 1,109.8 | 506.5 | Middle |
| LWJB | Wulong | 31,765 | 1,238.4 | 694.0 | Weak |
| QXGW | Qixingguan | 2,999 | 886.2 | 397.9 | Strong |





| DTW | Dongtou | 6,197 | 1,136.8 | 538.5 | Middle |
| YWW | Yongwei | 13,045 | 1,296.6 | 706.0 | Non-karst |
| TDW | Tongdao | 3,784 | 1,517.3 | 651.0 | Non-karst |

Figure 1: An overview of (a) the study area, and (b)the karstification degree in the study area

## 2.2 Data

90 A total of 7 hydrological stations and 19 meteorological stations were involved in this study from 1957 to 2008. The daily precipitation data were obtained from China Meteorological Data Service Centre (http://data.cma.cn/en), and the areal precipitation of each basin was calculated using the Thiessen polygon method. The daily flow data were provided by the Information Centre of the Ministry of Water Resources. To avoid the influence of upstream flow on runoff in the middle and lower reaches of the watershed, the daily flow in the MWJB and LWJB need to be reduced in proportion to their upstream

95 inflows:

$$Q_{i,r} = Q_{i,r} \times \frac{\overline{Q_o - Q_{o,u}}}{\overline{Q_o}}, \tag{1}$$



Where $Q_{i,r}$ is the reduced flow on day $i$, $Q_{i,r}$ is the original flow on day $i$, $\overline{Q_o}$ is the mean value of the original flow, and $\overline{Q_{o,u}}$ is the mean value of the original flow at the upstream station. The extent of karst and development in Southwest China (Fig. 1b) is derived from the *Hydrogeological Record of Guizhou Province* (Han and Jin, 1996).

## 3 Methodology

### 3.1 Drought identification and characterization

Here we used the threshold level method to identify both meteorological and hydrological drought events. As suggested in literature (Rangecroft et al., 2019), the threshold value for each day of the year can be generally set as the 80th percentile value among all years. This rule was applied to both meteorological and hydrological drought identifications. The obtained threshold values were further smoothed using a 30-day moving average method (Fig. 2a). Also, the 30-day moving averaging was applied to the original hydrological and meteorological timeseries. Based on this, the threshold-based drought indices were calculated as the differences of the moving averaged meteorological/hydrological timeseries and the corresponding threshold values.

The obtained threshold-based drought indices were then screened, and periods with values below zero were identified as drought events (Fig. 2b). To eliminate minor and interdependent droughts (Kim et al., 2011; Van Loon and Van Lanen, 2012), drought events with a duration of less than 15 days were excluded from the analysis, and adjunct events with an interval period less than 10 days were merged as a single event. Typical drought characteristics were also defined and quantified, including drought frequency, drought duration, and drought deficit volume.





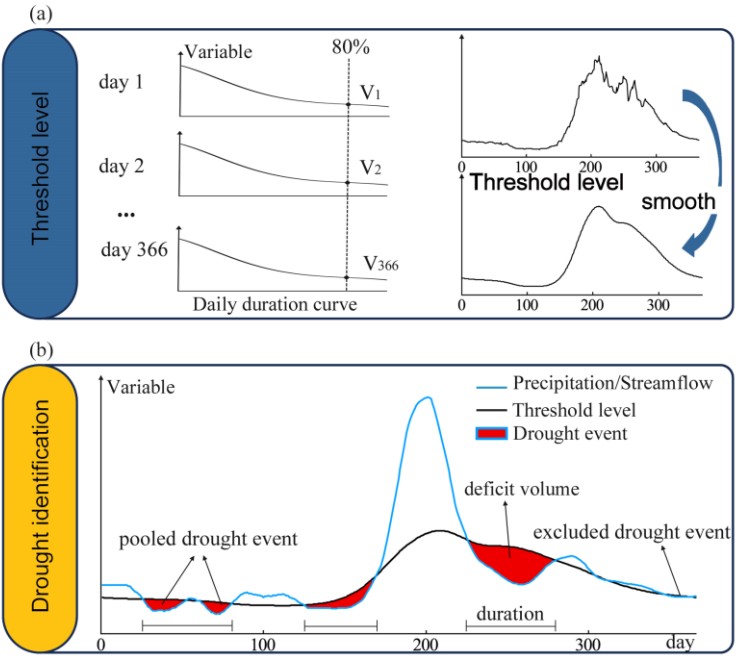

**Figure 2: Illustrations of Threshold level method with dynamic threshold (80th percentile of daily duration curve, smoothed by 30-day moving average) for meteorological and hydrological variable (precipitation and streamflow), including an illustration of pooling/excluding method and drought characteristics (duration and deficit volume)**

## 3.2 Calculations of the drought propagation time and the drought lag time

The common coefficient analysis method of Pearson's correlation coefficient (PCC) has been widely used to calculate the drought propagation temporal features (Guo et al., 2020; Tang et al., 2021; Zhao et al., 2016). In the literature, the PCC values (referred to here as PCC-A) were calculated between the meteorological drought index series (at different accumulative periods, e.g., ranging from 1 to 90 days in this study) and the hydrological drought index series. The accumulative period that achieves the maximum PCC-A is referred to as the drought propagation time ($T_P$) (Huang et al., 2015; Xu et al., 2019; Zhou et al., 2021c). A longer $T_P$ implies a slower response of streamflow to precipitation (Barker et al., 2016; Vicente-Serrano and López-Moreno, 2005). The drought lag time ($T_L$) is calculated similarly, but based on the delay period (ranging from 1 to 90 days in this study) of the meteorological drought index that achieves the maximum PCC-D (the PCC values between the meteorological drought index series at different delay periods and the hydrological drought index series) (Wu et al., 2021; Zhang et al., 2023).

As mentioned above, here we proposed the drought-period method to obtain $T_P$ and $T_L$ by using only the informative drought-period data as detected by the threshold level method in Section 3.1. Firstly, the recorded precipitation(streamflow) variables are calculated as a 30-day sliding average, and the meteorological and hydrological threshold level is subtracted to obtain





drought index. The meteorological drought index obtained is respectively accumulated for days (1,2, 3, …, 90 days) to obtain

the meteorological drought index series at different accumulative periods. Similarly, the meteorological drought index series

at different delay periods is obtained by considering different delay days (1,2, 3, …,90 days). Then, to achieves this, for each

of the detected hydrological drought events, we firstly obtained all PCC-A and PCC-D (for determining $T_P$ and $T_L$, respectively,

135    Fig. 3). Then, for each of the accumulative or delay days (ranging from 1 to 90 days), the average PCC-A and average PCC-

D among all events are calculated, and the corresponding accumulative and delay days of the maximum average PCC-A and

PCC-D are referred to as $T_P$ and $T_L$, respectively.

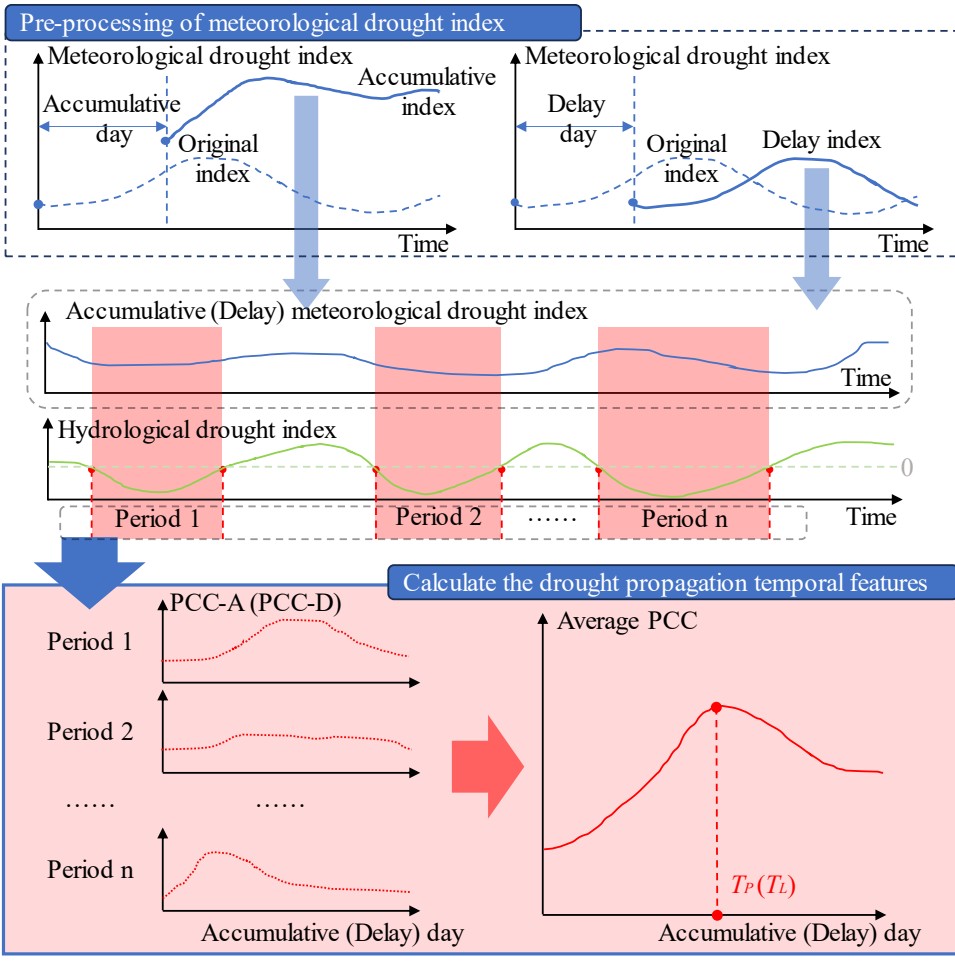

**Figure 3:** **Schematic diagram of the drought-period method. The red dashed lines show the PCC (PCC-A and PCC-D) between the**

140    **meteorological drought index at different accumulative(delay) periods and the hydrological drought index in each hydrological**

**drought period.**





### 3.3 Validation and evaluation of the new method on characterizing drought propagation

In general, the determined drought propagation features ($T_P$ and $T_L$) can be used to predict hydrological drought events that are triggered by the observed meteorological droughts. Typically, $T_L$ and $T_P$ are added to the start and the end times, respectively, of the meteorological drought event, obtaining the duration of the predicted hydrological drought event. As such, we can categorize them into four modes by comparing the timing and duration of the predicted and the observed hydrological drought events (Fig. 4a): (1) the drought do not occur in hydrological drought indices (Mode 1), (2) the former begins and ends both earlier than the latter (Mode 2), (3) the former totally envelopes the latter (i.e., starting earlier and ending later, Mode 3a), or inversely (i.e., starting later but ending earlier, Mode 3b), and (4) the former begins and ends later than the latter (Mode 4). In particular, Mode 2 and Mode 3 are referred to valid cases, because they represent that the meteorological droughts contribute to the occurrence and development of the hydrological drought events. Conversely, Mode 1 and Mode 4 are referred to as invalid modes due to the unrelated drought events.

Drought deficit volume is an important indicator in practical drought monitoring and drought prediction. Based on the duration of the hydrological drought event, the deficit volume ($V$) is quantified as the sum of the observed hydrological drought indices (i.e., the area below zero as shown in Fig. 4b).

$$V = \sum_{t=T_{begin}}^{T_{end}} I_t \quad (I_t < 0) \tag{2}$$

where $T_{begin}$ and $T_{end}$ is the start and end time of the hydrological drought event, respectively. $I_t$ are the observed hydrological drought indices at $t$ day.

In addition, we further evaluated the effectiveness of the proposed method by comparing to traditional method that is based on the full sequences of the drought indices. Particularly for the indicator of drought deficit volume. (Fang et al., 2020) and (Zhou et al., 2021c) stated that it is reasonable to further determine the rational $T_L$ and $T_P$ by comparing the effectiveness of diverse times in monitoring the streamflow deficits. In this study, we also paid attention to the drought deficit volume. As shown in Fig.4b, we firstly identified the valid meteorological drought events using the $T_L$ and $T_P$ obtained by the PCC methods for both the new drought-period method and the traditional full-sequence method, and then, compared the effectiveness of monitoring hydrological drought deficit volume. As follows:

$$\Delta E = E_N - E_T \tag{3}$$

$$E_N = \frac{\overline{V_N}}{\overline{V_O}} = \frac{(\sum_{i=1}^{n} V_{N,i})/n}{(\sum_{i=1}^{n} V_{o,i})/n} \tag{4}$$

$$E_T = \frac{\overline{V_T}}{\overline{V_O}} = \frac{(\sum_{i=1}^{m} V_{T,i})/m}{(\sum_{i=1}^{m} V_{o,i})/m} \tag{5}$$

Where $E_N$ and $E_T$ are the effectiveness of the new and traditional methods, respectively, in monitoring drought deficit volume, and their values indicate the ratio of the monitored drought deficit volume to the observed deficit volume. They range from 0

to 1. The closer the value is to 1, the more drought deficit volume of the observed hydrological drought can be monitored, so the more effective the method is. The $\Delta E$ represents the difference between the $E_N$ and $E_T$. When the value is positive, it indicates that the new method is more effective than the traditional one. $\overline{V_N}$, $\overline{V_T}$ are the average deficit volumes for all predicted hydrological drought events based on the new and traditional methods, respectively, and $\overline{V_O}$ is the average deficit volume for all correlated observed hydrological drought events. $n$ and $m$ are the number of the correlated observed hydrological drought events based on the new and traditional methods, respectively.

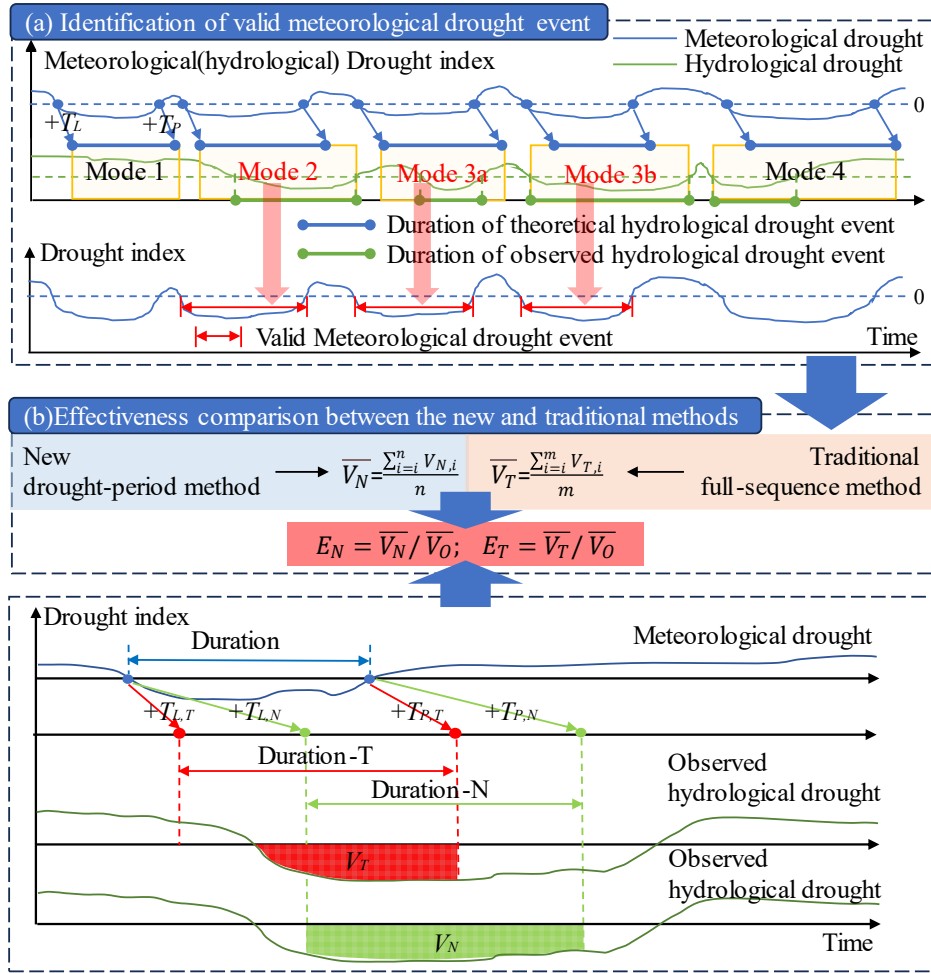

**Figure 4: Schematic diagram for the evaluation of new and traditional methods. The values of $E_N$ and $E_T$ indicate the effectiveness of the new and traditional methods, respectively, in monitoring drought deficit volume. $V_N$, $V_T$ are the deficit volume for predicted hydrological drought events based on the new and traditional methods, respectively. $T_{L,T}$ and $T_{P,T}$ are drought lag time and propagation time obtained by the traditional method. Similarly, $T_{L,N}$ and $T_{P,N}$ are drought lag time and propagation time obtained**





by the new method, respectively. Duration-T and duration-N are durations of theoretical hydrological drought event based on the traditional and the new methods, respectively.

## 4 Results & discussion

### 4.1 Characterization of drought events

Meteorological and hydrological drought events were firstly identified in all basins, with the characteristics of drought frequencies, durations and deficit volumes summarized in Fig. 5. The average frequencies of meteorological and hydrological droughts among all basins were 2.0 and 1.4 times per year, respectively (Fig. 5b), with the former ranging from 1.8 to 2.2 times per year and the later from 0.9 to 1.5 times per year (Fig. 5a). A greater difference in drought frequency can be found in hydrological drought (Fig. 5b, with the variance $\sigma$ as of 0.045) than in meteorological drought (Fig. 5b, with $\sigma$ as of 0.025) between watersheds. And the direct cause of this phenomenon is that the lowest frequency of hydrological drought at QXGW (0.9 times per year), which may be due to the low drought propagation rate in karst region (Shi et al., 2024). Furthermore, there are also different shifting trends in drought frequency observed between the watersheds (Fig. 5a). Specifically, the YWW exhibits a lower frequency of meteorological droughts compared to the DTW (1.8 < 2.2 times per year) but a higher hydrological drought frequency (1.5 > 1.3 times per year). The above results show that the frequencies of hydrological drought vary among watersheds with similar meteorological drought conditions. This further indicates that meteorological factors alone are not the sole contributors to hydrological droughts. It is crucial to consider the influence of other factors such as watershed characteristics as well as human activities when assessing the factors affecting hydrological droughts. In Fig. 5c and Fig. 5d, we can see than the durations of the meteorological drought were relatively similar across watersheds (no significant difference, with the $p$ as of 0.958 for the Kruskal-Wallis H Test), primarily concentrated between 20 and 23 days (25th to 75th percentile). On the other hand, the durations of hydrological drought are more dispersed (with the p as of 0. 145<0.958) and mainly concentrated in the range of 23 to 66 days (25th to 75th percentile). Similarly, the differences in durations between hydrological drought and meteorological drought vary across each watershed, and factors such as watershed characteristics and human activities may contribute to this difference. In contrast, there are significant differences in the meteorological drought deficit volumes among these watersheds (Fig. 5f, with a $p$ as of 8.50E-5), and these differences are related to precipitation. The highest average deficit volume of meteorological drought (Fig. 5e, 22.79 mm) is observed in TDW, which has the highest annual average precipitation (1517.3 mm, as shown in Table 1). In contrast, QXGW, which has the smallest annual average precipitation (886.2 mm), has the smallest mean deficit volume of meteorological drought (12.30 mm). However, when focusing on the deficit volume of the hydrological drought, the differences among the watersheds are not significant (with the $p$ as of 0.522). Due to the different regulation and storage effect of each watershed, the quantitative relationship of hydrological drought deficit volume among the watersheds differs from that of meteorological drought deficit volume. Therefore, it is necessary to consider not only meteorological variables but also the regulation and storage effect of the watershed when attributing the hydrological drought deficit volume.





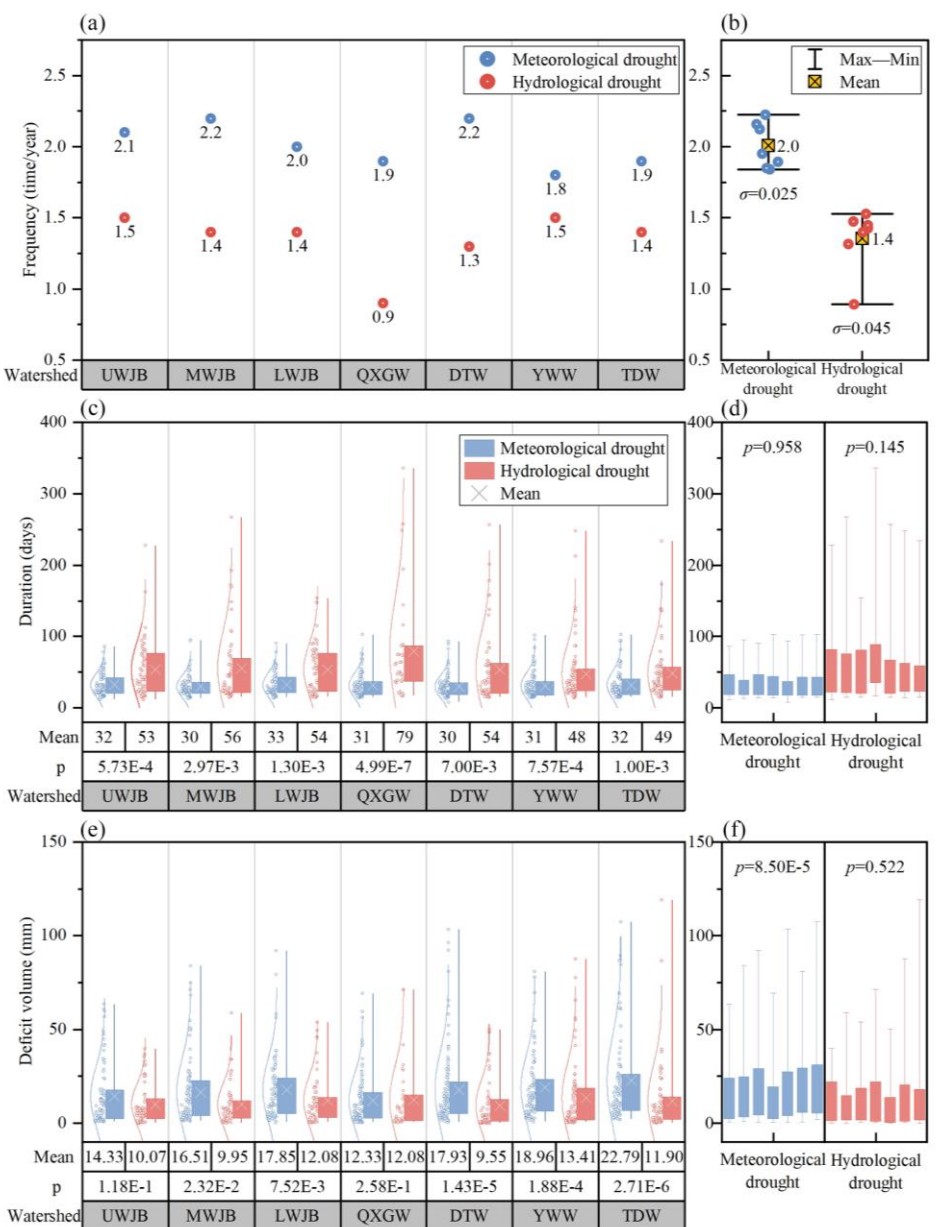

**Figure 5. Drought characteristics in all basins with meteorological drought in blue and hydrological drought in red; (a) and (b) shows the frequency of drought events, (c) and (d) shows the duration of drought events, (e) and (f) shows the deficit volume of drought events. σ represents the variance and p represents the Kruskal-Wallis H test value.**

Form Fig. 5a, the frequency of hydrological droughts in the UWJB is observed to be 1.5 times per year, whereas the frequency of meteorological droughts is higher, at 2.1 times per year. Moreover,the most intuitive result of Fig. 5a is that across all other watersheds, the frequency of hydrological droughts is lower than the frequency of meteorological droughts. This reveals that





the reduced frequencies during the propagation from meteorological to hydrological droughts. From Fig. 5c, the mean duration of the hydrological droughts is longer than the mean duration of meteorological droughts in all watersheds , and the all $p < 0.05$. Among them, the dissimilarity between the durations of the meteorological and hydrological droughts stands out prominently in QXGW (with the smallest $p = 4.99E-7$, and the comparison being 79>31 days). This finding suggests that QXGW is highly

prone to experiencing lengthened duration during the propagation, and a plausible explanation for this pattern may be the comparatively weaker drought recovery capacity in karst regions. Similarly, when we focus on the drought deficit volume(Fig. 5e), it can be observed that, except for UWJB and QXGW (with $p$ values of 1.18E-1 and 2.58E-1, respectively), the deficit volume of meteorological droughts in other watersheds is significantly larger than that of the hydrological droughts (with $p$ values of 2.32E-2, 7.52E-3, 1.43E-5, 1.884E-4, and 2.71E-6 in MWJB, LWJB, DTW, YWW, and TDW, respectively). It

becomes apparent that these two watersheds, UWJB and QXGW, are also characterized by a stronger karstification degree (Table 1). This further supports the notion that the regulation and storage capacity of UWJB and QXGW is weaker compared to other watersheds. Consequently, these two watersheds have a weaker capacity to mitigate the severity of drought impacts during the drought propagation.

The results mentioned above demonstrate that the frequency of hydrological droughts is lower than that of meteorological

droughts in all watersheds. Furthermore, the average duration of hydrological droughts is longer than that of meteorological droughts, and the average deficit volume of hydrological droughts is smaller than that of meteorological droughts in all watersheds. This outcome is consistent with previous studies, which have identified three characteristics of drought propagation. These include the reduced frequency (Fig. 5a), the lengthened duration (Fig. 5c), and the reduced deficit volume (Fig. 5e) during the propagation from meteorological to hydrological droughts (Gu et al., 2020). Additionally, the results also

indicate that catchment control is a non-negligible factor influencing drought propagation. Furthermore, during the periods of meteorological drought, the regulation and storage effect of the catchment means that not all such periods necessarily lead to hydrological drought event, resulting in a smaller frequency of hydrological drought events compared to meteorological droughts (Bevacqua et al., 2021; Chen et al., 2020; Shin et al., 2018; Zhou et al., 2021c). Moreover, even when hydrological drought event occurs, the antecedent water storage in its catchment mitigates the severity of drought, leading to a lower deficit

volume in hydrological drought event compared to meteorological drought event (Choi et al., 2021; Li et al., 2018). Of course, it is also due to the capacity for storage and regulation that there is a lag in the catchment response to precipitation events, leading to a slower recovery from hydrological droughts than from meteorological droughts, hence the longer duration of hydrological droughts (Gu et al., 2020; Wu et al., 2018, 2020).

**4.2 Drought propagation time and lag time**

Figure 6 shows the identification process of the $T_P$ and $T_L$ by using the new drought-period method for each watershed. It is observed that, overall, the average PCC initially increase and then decrease with the increase in cumulative or delay time in all watershed. Notably, the average PCC-D consistently reaches its peak before the average PCC-A, which results in all the $T_L$





being smaller than the $T_P$. For the watersheds UWJB, MWJB, LWJB, QXGW, DTW, YWW, and TDW, the $T_P$ are 17, 19, 16, 30, 26, 14, and 17 days, respectively. Correspondingly, the $T_L$ are 9, 8, 7, 17, 8, 5, and 9 days, respectively. Among these, the

maximum $T_P$ (30 days) and the maximum $T_L$ (17 days) both occur in QXGW. Conversely, the minimum $T_P$ is 14 days, as found in YWW, while TDW has the minimum $T_L$ (4 days). Previous studies have indicated that drought may be closely related to hydrometeorological factors such as precipitation and runoff (Han et al., 2019; Zhou et al., 2021b, c). (Zhou et al., 2021b) pointed out that the drought propagation time is correlated with the annual average precipitation and runoff, with regions having higher annual average precipitation and runoff experiencing shorter drought propagation times. Therefore, it is observed

that the watershed with the highest annual average precipitation is TDW (1517.3 mm), while YWW has the highest annual average runoff (706.0 mm). Correspondingly, the former has the shortest $T_L$ (4 days), and the latter has the shortest $T_P$ (14 days). Conversely, QXGW, which has the smallest annual average precipitation (886.2 mm) and runoff (397.9 mm), has the largest $T_P$ and $T_L$ (30 and 17 days, respectively). This such relationship has also been mentioned in (Zhou et al., 2021c), where regions with higher mean annual precipitation and higher runoff exhibit a faster response to changes in meteorological variables.

Additionally, we have also calculated $T_P$ and $T_L$ using the traditional method, the details of which are illustrated in Fig. 7. Similar to the results shown in Fig. 6, all $T_L$ are smaller than $T_P$. This is because the PCC increases with the accumulation of time or delay time, then decreases, and PCC-D always reaches its peak before PCC-A. The $T_L$ for UWJB, MWJB, LWJB, QXGW, DTW, YWW, and TDW watersheds are 6, 6, 5, 7, 4, 4, and 5 days, respectively, while the $T_P$ for these watersheds are 16, 16, 14, 20, 12, 13, and 14 days, respectively. Among these, QXGW has the maximum $T_P$ (20 days) as well as the

maximum $T_L$ (7 days). Conversely, the minimum $T_P$ (12 days) is observed in DTW, while the minimum $T_L$ is 4 days, found in both TDW and YWW.

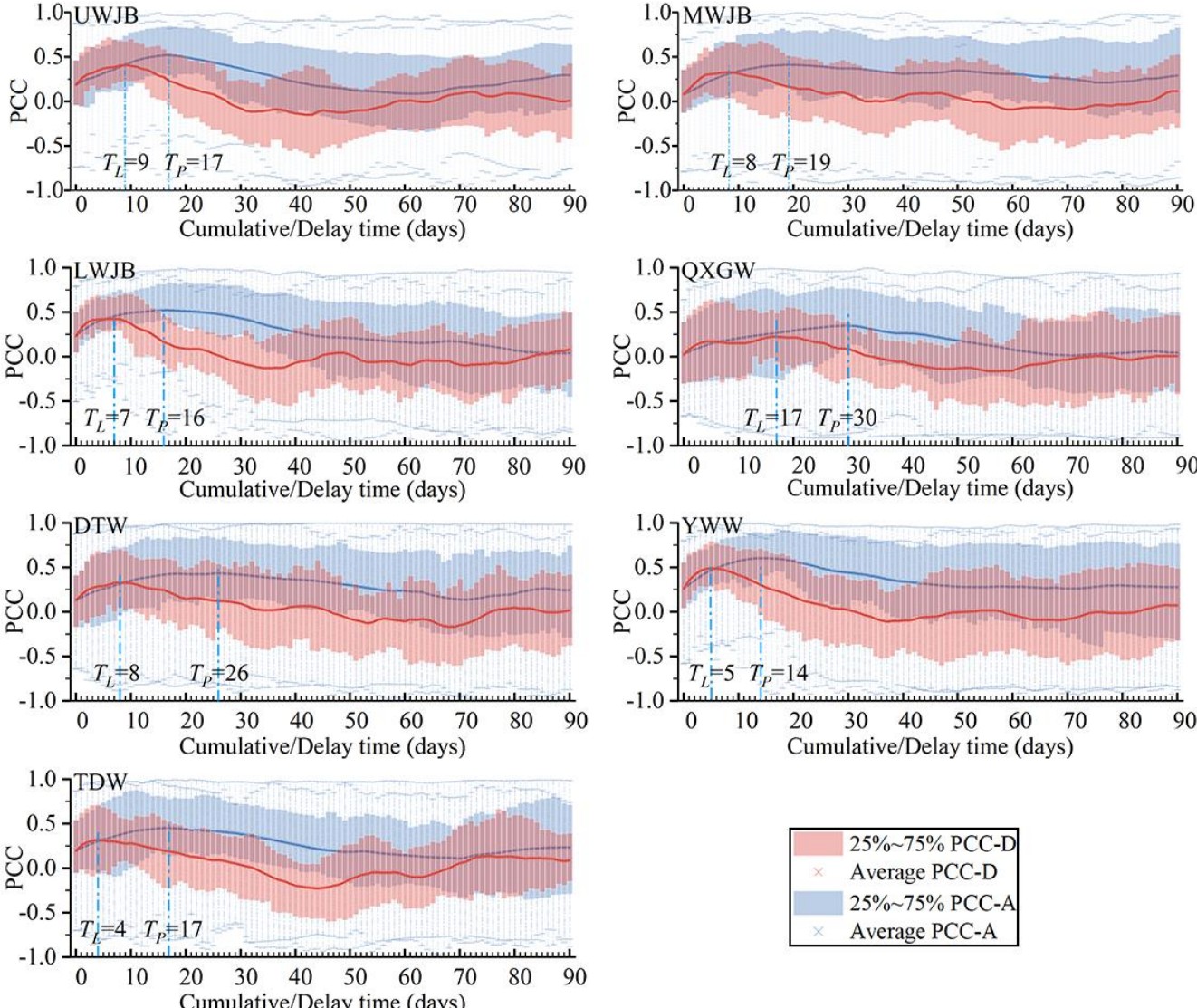

**Figure 6: The identification process of $T_P$ and $T_L$ in all basins by using the new drought-period method. The blue markers indicate the PCC-A (the PCC between hydrological drought index and accumulative meteorological drought index) in all drought periods.**
**The red markers indicate the PCC-D (the PCC between hydrological drought index and delay meteorological drought index) in all drought periods.**



**Figure 7: The $T_P$ and $T_L$ from meteorological to hydrological droughts for all basins obtained by using the traditional full-sequence method. The black solid line displays the PCC-A (the PCC between hydrological drought index and accumulative meteorological drought index). The red solid line displays the PCC-D (the PCC between hydrological drought index and delay meteorological drought index). The horizontal coordinate represents the accumulative time (delay time).**

To further compare the difference of the times obtained using the new and traditional methods in each watershed, Figure 8 shows the $T_P$ and $T_L$ obtained using the two methods. In Fig. 8a, the $T_P$ obtained using the new drought-period method (mean value as of 20 days) is greater than those obtained using the traditional full-sequence method (mean value as of 15 days) in all basins. The differences in $T_P$ for UWJB, MWJB, LWJB, QXGW, DTW, YWW and TDW are 1, 3, 2, 10, 14, 1 and 3 days,





respectively. From Fig. 8b, the $T_L$ obtained using the new method are also greater than those obtained using the traditional method for all basins except TDW (The difference in $T_L$ is -1 for TDW, and the differences in $T_L$ for UWJB, MWJB, LWJB, QXGW, DTW and YWW are 3, 2, 2, 10, 4 and 1 days, respectively). Previous studies have shown that there is a lag in the response of hydrological variables to meteorological variables, whether during drought or non-drought periods (Yang et al., 2022; Zhang et al., 2011). For example, the hydrological droughts start later than meteorological droughts during drought periods, Conversely, during non-drought periods, the flood peaks always occur later than the rainfall peaks (Gaál et al., 2012; Lu, 2009; Pendergrass et al., 2020). In Section 3.2, we know that the traditional full-sequence method is based on the overall hydrometeorological series, so the times obtained by the traditional method are the result of a combination of the lags in both drought and misinformative non-drought periods. However, we should focus more on the lag during drought periods when studying the propagation characteristics of hydrological droughts in response to meteorological droughts. Therefore, the new method that only considers information during drought periods should be used to obtain the times. Further, similar to the calculation of $T_L$ (see section 3.2), we quantified the lags (called lag time) of the hydrometeorological variables for drought and non-drought periods, and then compared them as shown in the Fig. 9. In all watersheds, there is a significant overall trend where the lag for drought periods is longer than that for non-drought periods (the mean lag time for drought periods is longer than that for non-drought periods in all watersheds, and all $p<0.05$). Due to the longer lag time for the drought periods, the $T_L$ obtained by considering only drought periods will always be longer than the $T_L$ obtained by considering the combined lag for both drought and non-drought periods. Therefore, it can be explained that the difference in these two lag times leads to the difference of $T_L$ (and $T_P$) obtained by the new and traditional methods.

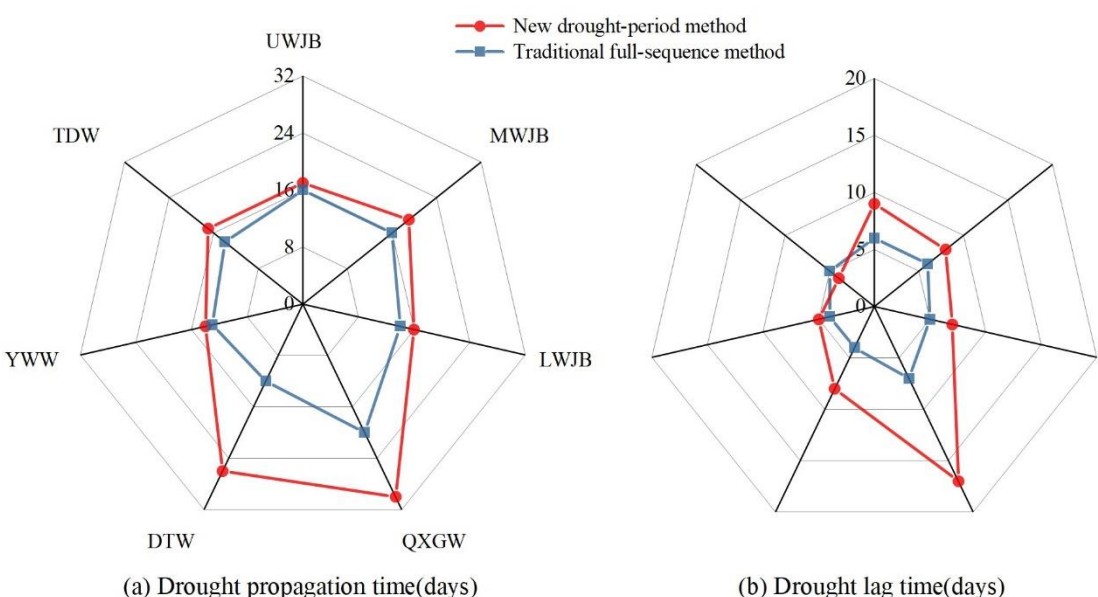

(a) Drought propagation time(days)          (b) Drought lag time(days)



**Figure 8: The $T_P$ (a) and the $T_L$ (b) in all basins obtained using both the new and traditional methods. The blue markers indicate the $T_P$ and $T_L$ obtained using the new drought-period method, and the red markers indicate the $T_P$ and $T_L$ obtained by using the traditional full-sequence method.**

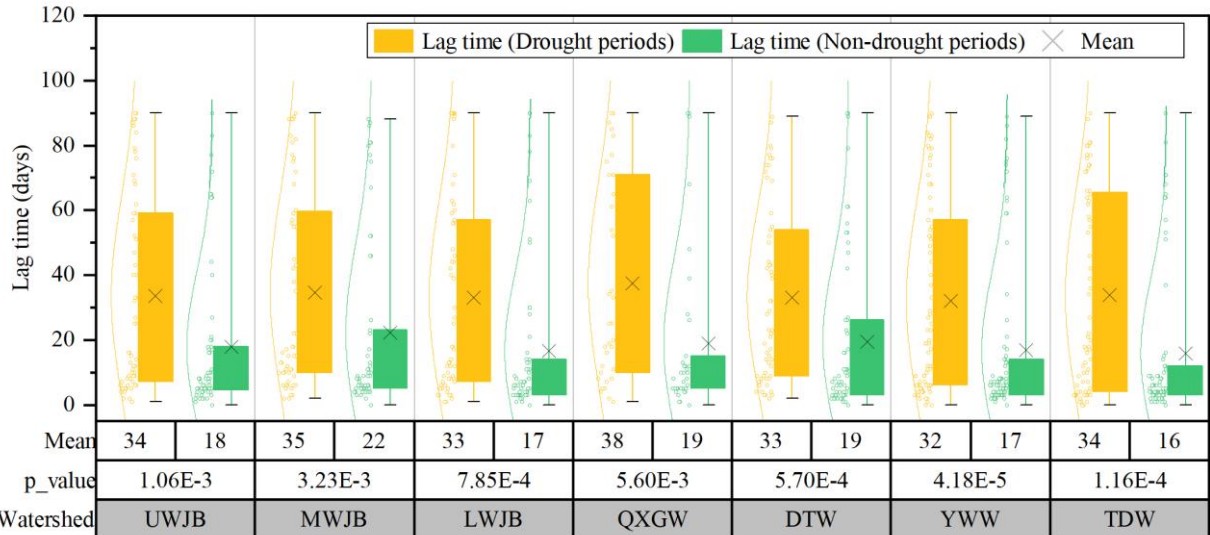

**Figure 9: The lag (quantified using the lag time, analogous to $T_L$) in all watersheds for drought periods and non-drought periods.**
**The lag time for drought periods is marked in yellow, and the lag time for non-drought periods is marked in green. The p_value represents the Kruskal-Wallis H test value.**

Furthermore, Figure 8 also shows that the larger differences in $T_P$ are found in QXGW (10 days) and DTW (14 days), which exhibit the stronger karstification degree. Similarly, the largest difference in $T_L$ (10 days) is also observed in QXGW. The appeal results indicate a correlation between the karstification degrees and the differences in $T_P$ (and $T_L$). The regions with
stronger karstification degree show larger differences in $T_P$ (and $T_L$). From the above analysis, we know that the annual average precipitation and runoff are also important factors affecting the drought propagation time. Therefore, in Fig. 9, we can see that QXGW has the longest lag time for the drought period (with an average lag time of 38 days), and its average lag time for the non-drought period is also longer (with an average lag time of 19 days). And this watershed has the smallest precipitation (886.2 mm) and runoff (397.9 mm). However, it would be expected that DTW, with more precipitation (1136.8 mm) and
runoff (538.5 mm) than QXGW, should have a shorter lag time for the non-drought period, but it is the same as that of QXGW (also with an average lag time of 19 days). A possible explanation is that the stronger karstification degree in QXGW further shortens the lag during the non-drought period. From the perspective of subsurface properties, the unique surface-groundwater hydrological connectivity in karst regions results in a lower storage capacity, and the permeable formations of karst systems contribute to frequent surface-groundwater interactions and fast routing processes (Chu et al., 2016; Wang et al., 2020).
Additionally, the weak water storage capacity due to shallow soils, as well as other factors, e.g., precipitation and evapotranspiration, makes the streamflow respond quickly to precipitation (Luo et al., 2023; Xu et al., 2020). Consequently,





in karst areas, the lag between meteorological and hydrological variables during the non-drought periods is further shortened. Therefore, it can be concluded that the difference in lag between drought and non-drought periods is influenced by both the precipitation and runoff, and by the karstification degree on the surface.

### 330   4.3 Effectiveness comparison between the two methods

Figure 10 illustrates the distributions of proportions and transformational relationships of the four drought propagation modes. The proportion of Mode 4 based on the new method is higher than that based on the traditional method (mean values as of 15.2% and 11.3% for the former and the latter, respectively). Conversely, the average proportion of Mode 2 based on the new method is 16.9%, which is less than that of the traditional method (13.1%). Through their transformational relationships, it can
be observed that partial meteorological drought events matched with Mode 2 based on the traditional method are transformed into meteorological drought events matched with other modes (Mode 3, and even Mode 4) based on the new method. Some of the proportion of Mode 4 based on the new method are transformed from the Mode 3 based on the traditional method. Consequently, a lower proportion of Mode 2 and a higher proportion of Mode 4 are observed compared to the traditional method. The direct cause of the above result is the differences in $T_P$ (and $T_L$) obtained by the new method in comparison to those obtained by the traditional method. As noted in Section 3.3 and Fig. 4, the determination of the Modes is primarily based
on the timing and duration of the theoretical and the observed hydrological drought events. In Section 4.3, the $T_P$ and $T_L$ obtained by the new method are consistently longer than those obtained by the traditional method. Consequently, the start and end times of theoretical hydrological drought events determined by the new method are later than those determined by the traditional method, leading to the above-mentioned transformational relationships. Next, the proportion of valid meteorological
drought events (the sum of the proportion of Mode 2 and Mode 3) based on the new method is 50.6% in UWJB, which is lower than that obtained based on the traditional method. A similar numerical relationship is also observed in other watersheds besides the TDW. For TDW, due to the small difference in $T_L$ between the two methods (the difference is -1 days in Fig. 8), the proportions based on the two methods are also similar (both 41.4%). In contrast, the largest difference in proportions between the two methods for DTW (9.4%) is explained by the largest difference in $T_P$ (14 days). However, the proportion of
valid meteorological drought events for QXGW is consistently the lowest whether using the new or the traditional methods (28.6% and 26.4%), which corresponds exactly to the lowest frequency of hydrological droughts (0.9 times per year), indicating that only a few meteorological drought events can influence the occurrence and development of hydrological drought events in this watershed.



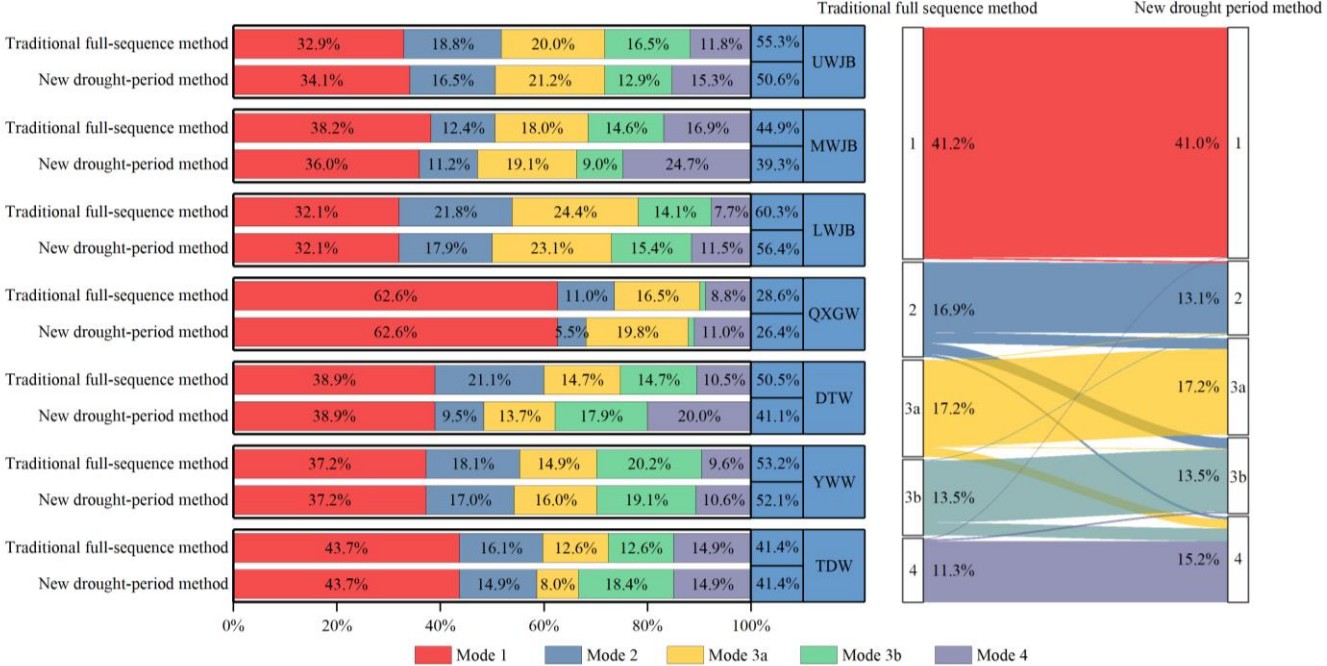

**Figure 10: The proportion of four drought propagation modes. The left panel shows the distributions of proportions of the four drought propagation modes, and the rightmost percentage represents the proportions of valid meteorological drought events for each basins, i.e., the sum of the proportions of Models 2 and 3. The figure on the right shows the transformational relationships between the proportions of the modes based on the traditional and new methods.**

Considering only the valid cases of Modes 2 and 3, the effectiveness of the $T_P$ and $T_L$ obtained using the two methods in monitoring the hydrological drought deficit volume is shown in Fig. 11. It is evident that the overall distribution of the effectiveness across the basins is consistent regardless of whether the new method or the old method is applied (QXGW<MWJB<UWJB<LWJB<YWW<DTW<TDW). Notably, the effectiveness of QXGW is the lowest, at 60.79% for the new method and 58.01% for the traditional method, respectively. This may be attributed to the longest duration of the hydrological droughts in this watershed (with a mean duration of 79 days), which coincides with multiple meteorological drought events. As shown in Fig. 12, there were two corresponding valid meteorological drought events (with drought durations from 2011/4/7 to 2011/6/22, and from 2011/8/21 to 2011/9/30, respectively) during the hydrological drought event with a long duration from 2011/5/12 to 2012/1/15. Its $E_T$ are 22.59% and 46.55% and the $E_N$ are 28.53% and 40.67%, both lower than its averages (58.01% and 60.79% in Fig. 11). The duration of the corresponding theoretical hydrological drought events was much shorter than that of the observed hydrological drought events because of the existence of the multiple meteorological drought events. Consequently, it leads to a low proportion of monitored deficit volume. In contrast, the TDW with fewer long duration hydrological drought events (Fig. 5d) had the highest effectiveness of 97.92% and 98.02%, respectively.





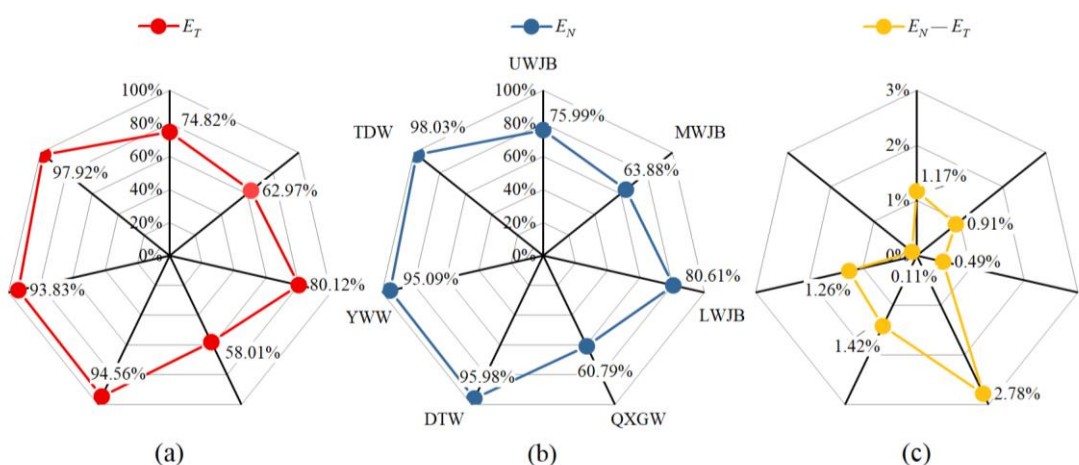

**Figure 11: the $E_T$ (a) and $E_N$ (b) and their difference (c) in all basins, and the $E_T$ and $E_N$ represent the effectiveness of the new and traditional approaches, respectively, in monitoring drought deficit volume.**

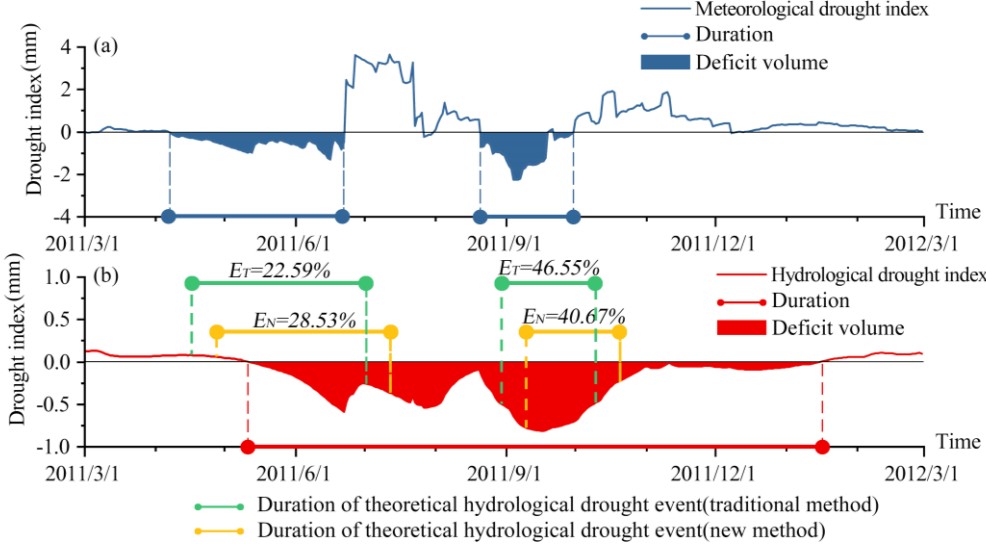

**Figure 12: The two corresponding valid meteorological drought events (with drought durations from 2011/4/7 to 2011/6/22, and from 2011/8/21 to 2011/9/30, respectively), occurred during the hydrological drought event with a long duration from 2011/5/12 to 2012/1/15.**

Furthermore, the $\Delta E$ for each basin is greater than 0, with an average of 1.16%, indicating that the new method is more effective as it can monitor more deficit volume than the old method. The $\Delta E$s for the three basins located on the main stream of the Wujiang River are 1.16% for UWJB, 0.90% for MWJB, and 0.47% for LWJB. Considering the karstification degrees in the Wujiang basin, more monitored deficit volume is observed in the upper reach (UWJB), which has a stronger karstification





degree, while in the lower reach (LWJB), where the karstification degree is weakest, $\Delta E$ is only 0.47%. Similarly, the
relationship between the karstification degree and $\Delta E$ is also observed in other basins, with QXGW (stronger karstification
degree) having the largest $\Delta E$, where the new method monitors 2.78% more deficit volume than the traditional method. In
contrast, $\Delta E$ is lowest in TDW in a non-karst region, with only 0.11%. As demonstrated in Section 4.2, the $T_P$ (and $T_L$) obtained
by the new method in karst regions is considerably larger than that obtained by the traditional method. For example, the $T_P$ in
QXGW (30 days) is greater than that obtained by the traditional method (20 days). In the case of the meteorological drought
event (from 2011/4/7 to 2011/6/22, as illustrated in Fig. 12), the larger $T_P$ (and $T_L$) obtained by the new method allows the
duration of the theoretical hydrological drought event to overlap more with the duration of the corresponding observed
hydrological drought event, thus monitoring more deficit volume (28.53% > 22.59%). However, in the non-karst region of
DTW, there is a smaller difference in $T_P$ (and $T_L$) between the new and traditional methods (the difference in $T_P$ is 3 days,
while the difference in $T_L$ is -1 day). Consequently, the increase in monitored deficit volume by the new method compared to
the traditional method is relatively minor (only 0.11%) in DTW.

Previous research by (Fang et al., 2020) highlighted that the capability of monitoring the maximum streamflow deficit volume
is the most important criterion for determining the reasonable drought propagation time, as it aligns with the practical
requirements of drought management. Similarly, the propagation time identified by (Zhou et al., 2021c) is associated with the
maximum streamflow deficit volume. Consequently, it is reasonable to determine the rational $T_P$ and $T_L$ by comparing the
effectiveness of diverse times in monitoring the greatest amount of deficit volume. Based on these above results, the new
method demonstrates better effectiveness in monitoring hydrological drought deficit volumes in studying the propagation of
meteorological drought to hydrological drought. Consequently, the $T_P$ and $T_L$ obtained using the new method are more
reasonable. Moreover, the new method shows the most significant improvement in effectiveness in karst areas compared to
non-karst areas, indicating that the enhancements made by the new method are more pronounced in karst regions.

## 4.4 Extensions and limitations

This study focuses on two types of drought: meteorological and hydrological drought. However, the propagation of
meteorological drought in the hydrological cycle is also relevant to other types of droughts: propagation to agricultural
drought(Li et al., 2022; Zhou et al., 2021a), and propagation to socioeconomic drought(Wang et al., 2023). Compared to the
longevity and availability of hydrological data, there are limitation of other types of data and the complexity of technology.
Because of this, there are fewer studies on other types of droughts and their propagation. Even so, the features and methods of
drought propagation between different types of drought are similar. (Xu et al., 2019) and (Huang et al., 2015) used the same
correlation analysis to study the propagation of meteorological drought to hydrological drought and agricultural drought,
respectively. (Wang et al., 2023), in their study on the propagation of meteorological drought to socioeconomic drought,
focused on the propagation threshold, as did other studies (Han et al., 2021; Wu et al., 2021). Therefore, the new drought-



period method proposed in this paper is also expected to be applied to the propagation of meteorological drought to other types of drought.

As obtaining data is challenging, the study only mentions the karstification degree as subsurface data, which is insufficient for characterizing subsurface features. Additionally, there are numerous factors related to karst that need to be considered. Future studies should provide a more detailed analysis of subsurface features. Moreover, due to the complex nonlinear dynamics of the hydrological system, drought propagation is usually controlled by multiple factors, and the combined effect of individual influences and multiple factors on the drought propagation remains an unclear mechanism, and further exploration of the effects of factors on drought propagation is worthy of further study. In addition, the Wujiang River Basin also has various water conservancy projects, and the regulating effect of these projects on the basin is not explicitly considered in this study even though human activities might have an important influence on drought propagation (Sutanto et al., 2024; Yang et al., 2024).

## 5 Conclusions

Drought propagation time and lag time are two important indicators for characterizing drought propagation, and thus, reasonably estimating these indicators are conducive to further understand the drought. Considering the shortcomings of the traditional full-sequence method, in this study, we propose the new drought-period method to obtain the drought propagation time and lag time. Then, the improved effectiveness of the new method is evaluated in both karst and non-karst regions. Our results can be summarized as follows:

1. The propagation from meteorological to hydrological drought is characterized by reduced frequency, lengthened duration, and reduced streamflow deficit volume. And the changes in these drought characteristics are influenced by meteorological variables as well as the regulation and storage effect of the watershed.

2. The $T_P$ (and $T_L$) obtained using the drought-period method is greater than that obtained using the traditional method. The larger differences in $T_P$ are found in QXGW (10 days) and DTW (14 days). Similarly, the largest difference in $T_L$ (10 days) is also observed in QXGW. This is influenced by both the precipitation and runoff, and by the karstification degree on the surface.。

3. The $T_P$ and $T_L$ obtained using the new method would be more reasonable to meet the practical requirements for drought monitoring, and it is also concluded that the improved effectiveness of the new method is better in karst regions than in non-karst regions. All basins have a ∆E greater than 0, range from 0.11-2.78%. For QXGW, where the karstification degree is stronger, it can monitor 2.78% more streamflow deficit volume with the drought-period method than with the traditional method. In contrast, the TDW (non-karst region) had the smallest improved effectiveness, only 0.11%.





The new drought-period method proposed in this paper and these findings can improve our understanding of the drought propagation features. With a view to better application for more accurate monitoring of streamflow deficit volume.

Furthermore, this can lead to an early hydrological drought warning and water resources management.

**Data availability**

All climate data are publicly available for download from the China Meteorological Data Service Centre(https://data.cma.cn/). Hydrological data are not available due to the Chinese policy.

**Author contribution:**

Conceptualization of the research was done by Han Tang, Peng Shi and Simin Qu. Data curation and formal analysis was done by Lanlan Zhao and Yiqun Sun. The original draft was written by Han Tang. Review and editing were done by Xiaoqing Yang, Peng Jiang and Qiongfang Li.

**Competing interests**

The authors declare that they have no conflict of interest.

**Acknowledgments**

We are grateful for the financial support from the National Natural Science Foundation of China (Grant No. 52179011, U2243229, and 52209015), Zhejiang Water Resources Science and Technology Program (Grant No. RA2202).

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
