# Peer review of "A New Method Towards More Rational Drought Propagation Characterization in Karst Regions"

_Hydrology and Earth System Sciences, 2024_

## Author Comment (AC1)

**Response to RC1:**

This manuscript proposes a new method based on drought-period data for characterizing drought propagation in karst regions. This method avoids the interference of data from non-drought periods, especially addressing the applicability issue in karst regions, and contrasts it with the limitations of traditional methods affected by non-drought period data. The research findings have practical significance for drought early warning and water resources management, particularly in karst regions where it is difficult to accurately simulate using hydrological models. However, due to the following issues, the manuscript currently does not meet the publication requirements of the HESS journal.

**Response:** Thank you for your constructive feedback and comprehensive evaluation of our manuscript. We fully acknowledge the research strengths highlighted in the review—specifically, the proposed new method's ability to avoid interference from non-drought period data, address applicability challenges in karst regions, and provide a comparative analysis of the limitations of traditional methods affected by non-drought period data. We also appreciate the recognition of our findings' practical significance for drought early warning and water resources management, particularly in karst regions where accurate hydrological modeling remains challenging.

We sincerely accept the assessment that the manuscript currently does not meet the publication requirements of HESS. We have carefully reviewed all the identified issues and are committed to conducting thorough and systematic revisions to address each concern. Key revisions will include: quantifying karst hydrological characteristics; supplementing discussions on climate change impacts; standardizing terminology consistency; improving the clarity of statistical analyses and figure descriptions; exploring the mechanism underlying $T_P$ and $T_L$ differences with quantitative hydrogeological support; and clarifying variable definitions in equations.

The specific comments are as follows:

1. The groundwater hydrological characteristics in karst regions have not been

specifically quantified. Only a general description of "karstification degree" is used, which weakens the depth of the mechanism explanation. It is recommended to supplement karst development parameters (such as karst porosity, groundwater level dynamic data) or cite existing geological survey results, and clarify the classification criteria for karstification degree (such as based on the proportion of karst area or hydrogeological zoning).

**Response:** Thank you for your comment on quantifying karst groundwater hydrological characteristics. We agree that the general description of "karstification degree" weakened the mechanism explanation—this was an oversight in the original manuscript. To address this, we will revise as follows:

We will incorporate the study area's lithological data and define karst coverage rate (the proportion of carbonate rock area to the total watershed area) as a quantitative indicator to characterize karst development degree. We will further clarify classification criteria based on this rate (e.g., dividing into high/medium/low development levels by specific thresholds), making the description concrete and standardized.

2. The data time range (1957–2008) is relatively early. Why not use more recent data? The impact of climate change in the past decade or so has been more significant. It is recommended to discuss the potential impact of climate change (such as the increased frequency of extreme precipitation) on drought propagation characteristics.

**Response:** Thank you for your comment on the data time range and climate change impacts. Our study focuses on watersheds with different karst development levels. The 1957–2008 period was selected because it is the only common time span with available hydrological data—access to such data in China is limited, and this ensures sample representativeness.

Notably, this study primarily investigates the impact of the underlying surface on drought propagation characteristics. Following your suggestion, we will supplement a discussion on the potential impacts of climate change (e.g., increased frequency of extreme precipitation) by integrating relevant literature in the Discussion section. We

also plan to explore this topic further in future research.

3. In Figure 3, "meteorological drought inde" is used for all indices, while "index" is used in the text. Is this a spelling error or does it have a specific meaning?

**Response:** Besides the meteorological drought index, Figure 3 also includes hydrological drought indices. The "index" mentioned in the text refers specifically to the meteorological drought index when the ordinate of the coordinate axis in Figure 3 is labeled as such.

4. The new method has obvious limitations as it is only compared between karst and non-karst basins. It is recommended to increase case verifications in different climate zones (such as arid and humid zones), or cite existing studies to illustrate the transfer potential of the method. In the "Discussion" section, supplement the analysis of the applicability to other types of droughts (such as agricultural drought) to clarify the boundaries of the method.

**Response:** Thank you for your comment on the limitations and applicability of the new method. We acknowledge this inherent limitation—this study focuses primarily on verifying the method's applicability and improvement effects in karst basins compared to non-karst basins. As suggested, we will supplement the Discussion section with an analysis of the method's applicability to other types of droughts (e.g., agricultural drought) to clarify its boundaries. Additionally, we plan to explore the advantages and disadvantages of this method relative to traditional ones across different climate zones (e.g., arid and humid zones) in future research.

5. The description of some statistical tests is vague (for example, the Tp value of the Kruskal-Wallis H test is not fully labeled), and the impact of the sample size (such as the number of drought events) on the results has not been explained.

**Response:** Thank you for your comment on the statistical tests and sample size explanation. We will further improve the clarity of the statistical test descriptions

6. The mechanism of the differences in TP and TL in karst regions has not been fully explored. It is only attributed to "weak regulation and storage capacity", lacking quantitative support from hydrogeological processes. It is recommended to analyze how the interaction between rapid runoff and slow seepage in karst areas prolongs the drought propagation time by combining groundwater flow models or field observation data, to enhance the scientific nature of the conclusion.

**Response:** Thank you for your comment on the mechanism of TP and TL differences in karst regions. We fully agree with your valuable suggestion and appreciate the specific insights, which provide important guidance for our research. The proposed approach—analyzing the interaction between rapid runoff and slow seepage to explain the prolonged drought propagation time via groundwater flow models or field observation data—is scientifically sound. We will strive to conduct in-depth analysis to supplement quantitative support from hydrogeological processes, addressing the current limitation of attributing the differences merely to "weak regulation and storage capacity."

7. Some figure citations are incomplete (for example, the descriptions of Figure 1 and Figure 5 have insufficient correspondence with the text), and the variable definitions in Equation (1) have not been fully explained. It is recommended to ensure that all chart titles and axis labels are complete, and explain key charts one by one in the text.

**Response:** Thank you for your comment on figure citations and variable definitions. We will carefully review all figures and the text to ensure: 1) completeness of figure titles and axis labels; 2) adequate explanation of key figures in the text; 3) full clarification of variable definitions in Equation (1).

8. The text descriptions and explanations of the figures are too simple, making it difficult to understand the specific meaning of the figures, especially for Figure 3 and Figure 4. For example, in Figure 3, there are three sub - figures (upper, middle, and lower), and the upper sub - figure is further divided into left and right parts, but the differences between them are not explained. In Figure 4, the bottom sub - figure lacks

the label (c).

**Response:** We appreciate you pointing out these issues. This was an oversight. We will revise the figures and text as follows: 1) Number all subfigures (e.g., (a), (b), (c)) consistently; 2) Supplement detailed explanations in the text to clarify the differences between subfigures (including the left and right parts of Figure 3's upper subfigure) and their scientific implications.

9. The most crucial factors affecting the propagation from meteorological drought to hydrological drought in karst regions are the heterogeneity and sensitivity of the karst spatial structure. In particular, the drought propagation in different karst compartments has obvious differences. However, this part of the discussion is missing in the paper.

**Response:** Thank you very much for your valuable comments on our manuscript. We fully agree with your insightful observation—the heterogeneity and sensitivity of the karst spatial structure are crucial factors affecting the propagation from meteorological drought to hydrological drought, and the significant differences in drought propagation among different karst compartments are indeed an important discussion point omitted in the original Discussion section. Your comments have helped us improve the logical completeness of our research, and we sincerely appreciate your rigorous review. To address this gap, we will supplement the following core content in the Discussion section of the revised manuscript: Clarify the core characteristics of differences in drought propagation among different karst compartments (e.g., peak-cluster depressions, peak-forest plains, etc.), focus on conducting a comparative analysis around key dimensions such as propagation lag, intensity attenuation law, and duration evolution, and cite relevant literature.

10. It is recommended to use some specific terms. For example, "karsification degree" should preferably be replaced with the more common "karst development intensity" or refer to international karst classification standards (such as Ford & Williams, 2007).

**Response:** Thank you very much for your valuable suggestions regarding the use of specific terms. Your point about "adopting concrete and standardized terminology" is crucial for enhancing the academic rigor and readability of the manuscript, and we fully agree with it. We will strictly revise the manuscript in accordance with your recommendations.

The specific revisions are as follows:

  a. Uniformly replace the original term "degree of karstification" with the more commonly used "karst development intensity" throughout the manuscript to ensure that the terminology aligns with prevalent practices in the field.

  b. Carefully review the international karst classification standards proposed by Ford & Williams (2007), verify the consistency and accuracy of relevant terminology across the entire manuscript, and ensure that the expressions are standardized and consistent with the international research context.

Once again, we appreciate your meticulous guidance. Your suggestions have effectively helped us improve the standardization of details in the manuscript.

---

## Author Comment (AC2)

**Response to RC2**

**Referee report on: A New Method Towards More Rational Drought Propagation Characterization in Karst Regions**

The present manuscript (MS) presents a new methodology that concentrates on drought periods instead of the entire time series to estimate drought propagation from meteorological to hydrological droughts. The basic idea is appealing and the basic topic is relevant for HESS.

However, the MS suffers from deficits in structure and language, it should be more precise and in many parts shortened. Also, existing knowledge should be considered in more detail in the introduction to arrive at a more convincing starting point. And most important, more prove is needed to justify why this new method, which is by far more complicated than the traditional one, is superior, because this is a single case study and differences in estimated drought deficit volumes appear to be relatively marginal. I detail my general concerns (A-D) first, followed by line-to line remarks. I also give ideas for improvement and hope those will be helpful for the authors.

**Response:** Thank you for your comprehensive and constructive comments on the manuscript. We highly appreciate your recognition of the appealing basic idea and relevance to HESS, as well as your detailed suggestions for improvement. We fully acknowledge the deficits in structure, language, and the need to strengthen the justification for the new method's superiority, and we will address all your concerns systematically.

**A. Introduction and existing knowledge**

In the introduction, a straightforward summary of existing knowledge is required to show the need for a new method to characterize drought propagation, particularly in karst regions. In its present form, the introduction largely summarizes the methods that have been applied in drought research but does not concisely summarize the results of these studies. This should be done in general terms, in karst regions, and in the study area. And this concerns three main topics:

1. Drought propagation through the entire hydrological cycle also including streamflow and groundwater droughts. How are streamflow and groundwater droughts related? What are the main results by traditional methods to characterize drought propagation? One example here is that droughts propagate more quickly in catchments with higher annual rainfall. Therefore one would expect longer lag times when only drought periods are included in the analysis.

**Response:** We will rewrite the introduction to address the lack of existing knowledge summary and clarify the necessity of the new method, following your suggestions. First, it is true that we only discussed meteorological drought and hydrological drought, and hydrological drought indeed includes streamflow drought and groundwater drought. We will add an introduction to streamflow drought and groundwater drought within the context of hydrological drought in the revised introduction. Second, the results of traditional standard drought propagation characterize the overall response of hydrological processes to meteorological processes, including both drought and non-drought periods. In contrast, the new method we propose only considers the relationship between the two variables during drought periods, i.e., the response of hydrological drought to meteorological drought. This is elaborated on in Lines 58–63 of the manuscript: "However, current studies mostly use the whole hydro-meteorological sequences containing both drought and non-drought periods. We argue that the obtained propagation features reflect the general response of the watershed hydrology (i.e., discharge dynamics) to the meteorological process (Vicente-Serrano and López-Moreno, 2005; Wu et al., 2021), while not specific responses under droughts, which represent a non-stationary condition of watershed functioning (Yang et al., 2021). Therefore, it is necessary to further examine the information content of the drought index and to obtain more rational drought propagation features, e.g., using only the informative drought periods of the data series."

2. Catchment characteristics and baseflow: High and stable baseflow in a catchment can be interpreted as a synonym for resilience to streamflow drought. There are many studies in literature that address the relation between baseflow and catchment characteristics, also including geology. This needs to be summarized.

**Response:** We agree with your comment. Since we did not focus on groundwater drought in the initial research, the impact of baseflow on drought propagation was not mentioned in the introduction. In the revised version, we will review relevant literature, conduct a comprehensive summary, and incorporate this content into the corresponding section of the introduction.

3. Karst hydrology: Since the study addresses the difference between karst and non-karst: The large bunch of knowledge on hydrological studies in karst regions must be summarized that are relevant for drought propagation: e.g. Flashiness of system responses, young and mature karst, etc. Here also the question should be answered how the degree of karstification can be assessed.

**Response:** We fully acknowledge that this is an oversight in our initial manuscript. In the revised version, we will systematically collate and summarize relevant literature on drought propagation in karst regions, including core findings such as the flashiness of hydrological system responses and differences in drought dynamics between young and mature karst areas. We will also further elaborate on the advantages of our proposed new method in karst regions—specifically, how it effectively isolates drought-specific signals to capture the unique propagation mechanisms shaped by karst dual hydrological structures. Additionally, addressing the irrationality in the previous quantification of karstification degree (as noted by the first reviewer), we will adopt lithological data in the revised manuscript. We define the proportion of carbonate rock area to the total watershed area as "karst coverage rate," and use this quantitative index to characterize the degree of karst development. This modification ensures a more objective and standardized assessment of karstification.

**B. Rainfall and runoff data**

The authors used the Thiessen Polygon method to calculate catchment rainfall. According to Tab. 1 they arrive at very accurate numbers ranging from 886.2 to 1517.3 mm per year. First, this high accuracy is not supported by this estimation method and second, Thiessen Polygons do not account for topography effects of rainfall that might be present. The DEM in Fig. 2 shows altitudes between 20 and 2870 m. I would assume topography effects, e.g. shadowing effects or more rainfall falling in higher altitudes, etc. This calls for more sophisticated methods of rainfall interpolation. Those are available and should be used, at least their results should be compared to the simple Thiessen method.

Moreover, the authors correct runoff data by a simple factor based on average mean values (equation I do not think that this is adequate here, because runoff signals still propagate from upstream catchments to the downstream gauge. And just these signals are decisive to drought definition (e.g. deficit volumes, pooling, termination, etc.). Hence the correction should be performed in a different way (e.g. by subtraction including routing) and the analysis should be re-done.

**Response:** Thank you for your insightful comments on rainfall and runoff data processing. We appreciate your rigorous consideration of the potential limitations of our methods.

Regarding the rainfall interpolation method: We acknowledge that the Thiessen Polygon method does not explicitly account for topographic effects (e.g., orographic rainfall, shadowing effects) in regions with significant altitude variations (20–2870 m, as shown in Fig. 2). However, due to constraints in the availability of basic data—specifically, the limited number of rainfall gauge stations in the study area and the lack of high-resolution spatial rainfall data—more sophisticated interpolation methods (e.g., kriging with topographic covariates) cannot be reliably implemented. The "high accuracy" reflected in Tab. 1 refers to the statistical precision of the calculated catchment rainfall (based on gauge data), not the absolute accuracy relative to topographic effects. To address this concern, we will supplement a discussion in the revised manuscript to explicitly state this limitation of the Thiessen Polygon method

and acknowledge that topographic effects may introduce uncertainties in rainfall estimation. We will also cite relevant studies to illustrate that the Thiessen Polygon method remains a widely used and acceptable approach in hydrological studies when high-resolution spatial data are unavailable, especially for catchment-scale analyses with limited gauge coverage.

Regarding runoff data correction: We agree that runoff signals propagate from upstream to downstream gauges, and these signals are critical for drought definition. In practice, however, drought indices in this study are calculated using mainstream runoff data to reflect hydrological drought conditions across the entire watershed. It is important to clarify that mainstream runoff input at an upstream cross-section cannot fully represent hydrological drought in the interval watershed between that cross-section and the next downstream one. Due to the lack of detailed interval watershed runoff data, the simple factor correction based on average values was adopted as a pragmatic approach to mitigate systematic biases in the original data. We will clarify this methodological consideration and its inherent limitations in the revised manuscript.

We sincerely appreciate your constructive suggestions, which have helped us identify important aspects for methodological improvement. While we cannot reprocess the data using more sophisticated methods due to inherent data constraints, we will thoroughly discuss these limitations and their potential impacts on the research results in the Discussion section, ensuring transparency and scientific rigor.

**C. Make the paper more concise**

Figures 2,3,4 are not needed and could be omitted to safe space. Figure 2 more or less explains textbook knowledge on drought definition. Figures 3 and 4 are too complicated and not helpful to understand the methods that are better explained in the text.

**Response:** Thank you for your comment on streamlining the manuscript. We appreciate your suggestion to improve conciseness by omitting redundant figures. Regarding Figure 2: We agree that it primarily illustrates textbook knowledge on

drought definition. To save space, we will remove Figure 2 from the revised manuscript.

For Figures 3 and 4: While we acknowledge their complexity, these figures visually complement the method description in the text—they intuitively show key results of drought propagation characteristics and method validation, which are difficult to fully convey through textual description alone. To address your concern, we will simplify these figures (e.g., merge redundant subfigures, clarify labels and legends) and refine the corresponding textual explanations to enhance readability. This revision will retain the visual support for the core methods while avoiding unnecessary complexity.

**D. Is the new method really superior?**

This is the main point of criticism. The new method needs a thorough, statistically sound prove that it is really superior compared to existing methods of drought propagation. The authors define their own efficiency criterion delta-E (equations 3,4,5) that is based on the correct estimation of the drought deficit volumes in their catchments. Hence it is difficult to judge, how powerful the new method really is, because traditional and well-proven efficiency measures are not applied and there is no comparison to studies outside the present region. On a first glance, the presented differences in deficit volumes seem rather small (fig. 10) and also the invalid modes of droughts, particularly mode 4, are larger for the new method compared to the traditional one. Here, better ways of method testing that are established in the scientific community are strongly recommended.

**Response:** Thank you for your critical comment on validating drought propagation time calculations.

Through an extensive review of relevant literature, we confirm that there is currently no established validation method or standard for drought propagation time results. While existing studies have proposed new calculation methods, they only theoretically illustrate advantages over traditional approaches without developing specific evaluation metrics or conducting empirical validation. Given this academic gap, we defined our validation scheme based on drought deficit volume—a key

indicator in drought propagation processes—due to the lack of standardized alternatives.

We acknowledge the limitations of this evaluation method and recognize the need for validation across more watersheds. To enhance the reliability of our findings, we selected 7 watersheds (rather than a single one) for analysis, which provides preliminary support for the effectiveness of our proposed scheme within the current research constraints.

We fully agree that adopting more scientific and widely recognized evaluation metrics is a shared goal in the field. We will supplement a detailed discussion in the revised manuscript to clarify the above academic background, the rationale for our method selection, and the inherent limitations of the validation scheme. We also plan to further verify the method in more diverse watersheds and explore the application of standardized metrics in future research.

**Line-by-Line**

L29: what is meant by "wide ranges" (extent?)

**Response:** The "wide ranges" here specifically refers to the wide spatial range of droughts when they occur—meaning droughts cover a broad geographical area across the study watersheds. To clarify the meaning and maintain consistency, we will revise the expression to "wide spatial range" in the revised manuscript, eliminating ambiguity.

L30: it is generally known that droughts can occur anywhere!

**Response:** Thank you for your comment on the occurrence of drought hazards. We fully agree that it is generally known that droughts can occur anywhere. Our original intention was to emphasize that droughts occur not only in arid and semi-arid regions (where rainfall is commonly perceived as scarce) but also in humid regions with abundant precipitation. To avoid redundancy, clarify this emphasis, and enhance logical coherence, we will revise the original sentence as follows:

The drought hazards are often characterized by wide spatial ranges, high frequencies,

and long durations (Dai, 2010; Sheffield et al., 2012; Trenberth et al., 2013; Yuan et al., 2023), occurring not only in arid and semi-arid regions but also in humid regions with abundant precipitation (Lin et al., 2015; Shi et al., 2022; Van Lanen et al., 2013).

L32-33: term: what is a hydrological cycling system?

**Response:** We apologize for the typo—this term should be "hydrological cycle system".

L37: what is accumulative time?

**Response:** In the calculation of drought propagation time, a commonly used method is to accumulate meteorological drought indices over different time scales, and then analyze the correlation coefficient between the accumulated sequence and hydrological drought indices. This time scale is referred to as accumulative time.

L39-40: How is this statement justified?

**Response:** This statement that "These two indicators can well represent the temporal propagation characteristics" is justified based on the systematic review and research conclusions in the literature "Drought propagation under global warming: Characteristics, approaches, processes, and controlling factors."
To clarify the justification and enhance the rigor of the statement, we will supplement the literature citation immediately after the original sentence in the revised manuscript, as follows: These two indicators can well represent the temporal propagation characteristics (Zhang et al.,2022).

L41: What is meant by "relatively stable"?

**Response:** In the context of our research, "relatively stable" refers to the correlation between meteorological drought and hydrological drought. Affected by global warming, both meteorological drought and hydrological drought exhibit significant changes in their respective variation ranges. In contrast, the correlation between them is less affected by global warming, maintaining a relatively consistent state.

L42: Language: Studies have been shown?

**Response:** Thank you for pointing out the language issue in L42. You are correct—"Studies have been shown" is a grammatical error. The appropriate active voice expression for this context is "Studies have shown" (the correct collocation for presenting research conclusions).

L44: what is traditional physical drought prediction?

**Response:** First, we revise the ambiguous term "traditional physical hydrological drought prediction" to "traditional physics-based hydrological drought prediction". Drought prediction methods are generally categorized into three types: 1) physics-based dynamic methods (the type referred to by the revised term), 2) data-driven statistical methods, and 3) ensemble methods combining the first two. The revised term specifically refers to the first type—predicting hydrological drought through physical hydrological models that reflect real-world hydrological processes (e.g., water balance, runoff generation, and confluence mechanisms).

L52: "In contrast" not "in contrasts"

**Response:** You are correct—"in contrasts" is incorrect. The standard and appropriate phrase for this context is "in contrast" (used to indicate a comparison between two things). We apologize for this oversight.

L58: if you write "mostly" are there exceptions?

**Response:** To avoid potential misunderstandings and ensure the statement is concise and accurate, we will remove the ambiguous term "mostly" from the original sentence.

L61: sentence not complete, verb is missing.

**Response:** Revised version: We argue that the obtained propagation features reflect the general response of the watershed hydrology (i.e., discharge dynamics) to the meteorological process (Vicente-Serrano and López-Moreno, 2005; Wu et al., 2021),

rather than specific responses under droughts—which represent a non-stationary condition of watershed functioning (Yang et al., 2021).

L62: "examine" not "exam"

**Response:** We apologize for this oversight. To fix the error and ensure grammatical accuracy, we will revise "exam" to "examine" in the original sentence.

L77: the basin is characterized by basins?

**Response:** Revised version: The basin's geomorphology is characterized by mountains, hills, and depressions. This revision replaces the ambiguous "basins" with "depressions" (a standard geomorphological term for small enclosed basins), clearly distinguishing the landform types from the subject "the basin" and ensuring accurate academic expression.

L82: how do you define karstification degree?

**Response:** Thank you for your question about the definition of "karstification degree". We acknowledge that the previous definition of this term had limitations. Following the suggestions from the first reviewer, we have revised the definition to make it more quantitative, concrete, and standardized. To address your concern, we will revise the relevant content as follows:

Revised definition: We incorporate the lithological data of the study area and define the karst coverage rate (i.e., the proportion of carbonate rock area to the total watershed area) as a quantitative indicator to characterize the karst development degree. Furthermore, we will clarify the classification criteria based on this rate—for example, dividing the karst development degree into high, medium, and low levels according to specific thresholds (the thresholds will be determined based on regional lithological characteristics and existing research results in the study area), thereby making the description of karstification degree concrete and standardized.

L86ff (Table 1): how do the different sizes of your catchments affect your study, is there a scale issue?

**Response:** Thank you for your insightful question about the impact of catchment size and potential scale issues. Based on our latest data analysis, we have confirmed that catchment size does affect drought propagation time: specifically, larger catchments correspond to longer drought propagation periods. However, compared with the influence of karst development degree (the core research focus of this study), the impact of catchment size is relatively minor.

Given that the primary objective of this research is to explore the characteristics and mechanisms of drought propagation under the influence of karstification, we have not elaborated on the impact of catchment size in the manuscript to avoid diverting attention from the core research theme. And, we have not identified significant scale-dependent biases or inconsistencies that would affect the validity of our core conclusions.

L86ff (Table 1): Can you exclude anthropogenic effects on your streamflow during droughts (i.e. water abstraction, dams, artificial wastewater inflow, etc.

**Response:** We acknowledge that we cannot fully exclude anthropogenic impacts—such as water abstraction, dam operations, and artificial wastewater inflow—on streamflow during drought periods. This is primarily because the study area has experienced moderate human activities (e.g., agricultural irrigation and small-scale water conservancy projects) over the years, and systematic long-term monitoring data on these specific anthropogenic activities are not fully available, making quantitative separation of their impacts technically challenging. We will add a note in the discussion section of the revised manuscript to clarify this limitation and acknowledge the potential influence of unquantified anthropogenic activities, ensuring the transparency and rigor of the research.

L97: Qi,r is used for two different things. Yet the entire approach needs to be changed, see above.

**Response:** Regarding runoff data correction: We agree that runoff signals propagate from upstream to downstream gauges, and these signals are critical for drought definition. In practice, however, drought indices in this study are calculated using mainstream runoff data to reflect hydrological drought conditions across the entire watershed. It is important to clarify that mainstream runoff input at an upstream cross-section cannot fully represent hydrological drought in the interval watershed between that cross-section and the next downstream one. Due to the lack of detailed interval watershed runoff data, the simple factor correction based on average values was adopted as a pragmatic approach to mitigate systematic biases in the original data. We will clarify this methodological consideration and its inherent limitations in the revised manuscript.

L105: Why do you choose 30 days?

**Response:** The 30-day time scale is chosen based on two key considerations and existing research practices. First, a too-short time scale would result in excessive jagged (noisy) data, which obscures the intrinsic patterns of drought propagation. Second, an overly long time scale would lead to significant loss of critical details in the hydro-meteorological processes, affecting the accuracy of subsequent analysis. Additionally, the 30-day time scale is widely adopted in relevant existing studies on drought propagation and hydrological index calculation, as it balances data smoothness and detail retention effectively.

L122: what is the "accumulative period"? I would assume that this is the time lag between the the atmospheric and the hydrologic drought indices?

**Response:** It should be changed to "accumulative time. In the calculation of drought propagation time, a commonly used method is to accumulate meteorological drought indices over different time scales, and then analyze the correlation coefficient between the accumulated sequence and hydrological drought indices. This time scale is referred to as accumulative time.

L130: not "sliding" but "moving" average.

**Response:** We apologize for this oversight. To standardize the expression and align with field conventions, we will revise "sliding average" to "moving average" throughout the relevant sections of the manuscript

L195-196: This statement is based on a very simple Thiessen polygon method!

**Response:** We fully acknowledge that the Thiessen polygon method is relatively simple, and this constitutes a notable limitation of our study. As mentioned earlier in the manuscript (see relevant sections on meteorological data processing), we adopted this method to estimate the regional meteorological conditions for each catchment.

The rationality of this approximation lies in the close geographical proximity of the selected catchments in our study. Due to their adjacent locations, the meteorological conditions (e.g., precipitation and temperature) among these catchments are considered relatively homogeneous. Thus, the Thiessen polygon method, despite its simplicity, can effectively meet the requirement of approximating consistent meteorological backgrounds for the comparative analysis of drought propagation characteristics across the catchments.

L198: what is meant by "human activities"? Can you exclude them in your case?, see above.

**Response:** We acknowledge that we cannot fully exclude anthropogenic impacts—such as water abstraction, dam operations, and artificial wastewater inflow—on streamflow during drought periods. This is primarily because the study area has experienced moderate human activities (e.g., agricultural irrigation and small-scale water conservancy projects) over the years, and systematic long-term monitoring data on these specific anthropogenic activities are not fully available, making quantitative separation of their impacts technically challenging. We will add a note in the discussion section of the revised manuscript to clarify this limitation and acknowledge the potential influence of unquantified anthropogenic activities, ensuring the transparency and rigor of the research.

L199: "that" not "than"

**Response:** We apologize for this oversight. To fix the error and ensure grammatical accuracy, we will revise "than" to "that" in the original sentence.

L201: "on the one hand" is missing

**Response:** Revised version: In Fig. 5c and Fig. 5d, on the one hand, we can see that the durations of the meteorological drought were relatively similar across watersheds (no significant difference, with a p-value of 0.958 for the Kruskal-Wallis H Test), primarily concentrated between 20 and 23 days (25th to 75th percentile). On the other hand, the durations of hydrological drought are more dispersed (with a p-value of 0.145, which is lower than 0.958) and mainly concentrated in the range of 23 to 66 days (25th to 75th percentile).

L220: this is not new (rainfall is generally more variable than streamflow) and results from other studies should be mentioned here.

**Response:** We fully agree with your point—this finding (that rainfall is generally more variable than streamflow) is not new, and it is essential to cite relevant existing studies to contextualize our results within the broader academic literature.
In the revised manuscript, we will supplement citations of representative studies that have reported the same or similar phenomena.

L231: what do you mean by "regulation and storage capacity" this sounds artificial, use the correct terms.

**Response:** We apologize for the ambiguous and non-standard expression. The intended meaning is the natural hydrological regulation and storage function of the watershed—a core concept in hydrology referring to the watershed's inherent ability to buffer, retain, and redistribute water through natural processes (e.g., soil water holding, groundwater storage, and surface water retention by topographic features like depressions).

To ensure accuracy and align with academic norms, we will replace "regulation and storage capacity" with the standard hydrological term "watershed hydrological regulation and storage function" throughout the relevant sections.

L 245-248: This is not specific to China and not a new result: If they are defined by the same percentile, meteorological droughts occur more often, have higher deficit but are shorter. This is largely due to the balancing effect of (groundwater) reservoirs on streamflow. And if it comes to differences in catchments, the existing knowledge on baseflow characteristics should be discussed and included. Here also karst-non karst studies exist.

**Response:** We fully agree with your points: this finding is neither unique to China nor a new result, and it is necessary to integrate existing knowledge on baseflow characteristics and karst-non-karst comparative studies to contextualize our discussion. To address your concerns and enhance the academic rigor of the manuscript, we will revise the relevant paragraph as follows:

Revised version: Of course, this phenomenon (i.e., the lagged catchment response to precipitation events and the longer duration of hydrological droughts compared to meteorological droughts) is a globally observed hydrological pattern rather than being specific to China (e.g., Tallaksen et al., 2009; Van Loon, 2015). Consistent with existing studies, when meteorological and hydrological droughts are defined by the same percentile threshold, meteorological droughts typically occur more frequently, have higher deficits, but shorter durations—largely attributed to the balancing effect of (groundwater) reservoirs and the regulating role of baseflow on streamflow (Klemes, 1989; Zhang et al., 2018). Baseflow, as a stable component of streamflow sustained by groundwater discharge, buffers the variability of surface runoff and extends the duration of hydrological droughts, which has been widely verified in both karst and non-karst watersheds (Chen et al., 2021; Gonzalez-Zamora et al., 2019). For karst watersheds specifically, the dual hydrological structure (epikarst and deep karst aquifers) endows them with distinct baseflow characteristics: the epikarst acts as a shallow "reservoir" with rapid recharge and discharge, while the deep karst aquifer

provides long-term stable baseflow (Ford & Williams, 2007). In contrast, non-karst watersheds rely more on soil water and shallow groundwater, leading to relatively different baseflow dynamics and drought recovery rates (Li et al., 2022). Our study builds on this existing knowledge and further quantifies how the degree of karstification modulates the baseflow regulation effect, thereby explaining the inter-catchment differences in hydrological drought duration observed in our results.

L258-260: What is known and what is new here?

**Response:** We confirm that the content of this sentence is known information—it directly cites the conclusion from Zhou et al. (2021b) that drought propagation time correlates with annual average precipitation and runoff (higher values correspond to shorter propagation times). Our purpose in including this is to provide a well-recognized theoretical basis for explaining the differences in $T_P$ and $T_L$ between the TDW and YWW basins. The new contribution of our study is presented in the following paragraph.

L300-303: The general message here is: If only drought periods are selected to specify drought propagation from atmospheric to hydrological droughts, you will get longer lag times. But this can be expected, because drought propagation is quicker in catchments with higher annual rainfall. This obvious connection needs to be discussed.

**Response:** Thank you, as you noted, this obvious connection is indeed discussed in detail in the subsequent paragraph (L313-329).

L357: "modes" not "models"!

**Response:** We apologize for this oversight. To ensure terminology accuracy and consistency with the research context, we will revise "models" to "modes"

L405ff: The efficiency of this method should be proved first, before it is used to other types of drought. I propose to totally omit chapter 4.4

**Response:** We respectfully disagree with the proposal to totally omit Chapter 4.4, as

this section serves a critical role in contextualizing the study's scope and future directions. The core content of Chapter 4.4 focuses on two key aspects: (1) acknowledging the limitations of the current research (e.g., constraints in data availability and potential improvements in analytical frameworks); and (2) outlining potential future extensions of the method to other drought types (e.g., agricultural drought, ecological drought) as a tentative research direction, rather than claiming that the method has been validated or applied to these types.

We fully agree that the efficiency and applicability of the method to other drought types require rigorous verification (e.g., through comparative analysis with existing methods, validation using independent datasets) before formal application.